# Influence of *Pediococcus pentosaceus* GT001 on Performance, Meat Quality, Immune Function, Antioxidant and Cecum Microbial in Broiler Chickens Challenged by *Salmonella typhimurium*

**DOI:** 10.3390/ani14111676

**Published:** 2024-06-04

**Authors:** Gifty Ziema Bumbie, Leonardo Abormegah, Peter Asiedu, Akua Durowaa Oduro-Owusu, Achiamaa Asafu-Adjaye Koranteng, Kwabena Owusu Ansah, Vida Korkor Lamptey, Chen Chen, Taha Mohamed Mohamed, Zhiru Tang

**Affiliations:** 1Laboratory for Bio-Feed and Molecular Nutrition, College of Animal Science and Technology, Southwest University, Chongqing 400715, China; giftyziema@gmail.com (G.Z.B.); m15587125971@163.com (C.C.); 2Council for Scientific and Industrial Research, Animal Research Institute, Accra 20, Ghana; davinciabor@gmail.com (L.A.); sikafuturoa111@gmail.com (A.D.O.-O.); achiamaa1@yahoo.com (A.A.-A.K.); ojake6@gmail.com (K.O.A.); kortei_1999@yahoo.com (V.K.L.); 3Department of Animal Production and Health, School of Agricultural and Technology, University of Energy and Natural Resources, Sunyani 214, Ghana; pierro605@gmail.com; 4Department of Animal and Fish Production, Faculty of Agriculture (Saba Basha), Alexandria University, Alexandria 21531, Egypt; tahaabdelameed@alexu.edu.eg

**Keywords:** poultry, broiler, *Salmonella typhimurium*, probiotic, *Pediococcus pentosaceus*

## Abstract

**Simple Summary:**

*Salmonellosis* affects the poultry industry globally and is a major cause of bacterial food poisoning in humans. A *Salmonella* infection in broiler chickens can result in lower growth rates, decreased feed efficiency and higher mortality rates. The inclusion of probiotics in chicken diets enhances their intestinal microbiota and enhances different patterns of cytokine production in response to *Salmonella* infection. The increasing demand from consumers for chicken products devoid of antibiotics has increased the need to identify alternatives to antibiotics for the management of *Salmonella* infection, colonization, and carcass contamination in poultry. Through the process of competitive exclusion, probiotics have been shown to inhibit the colonization of the gut by harmful bacteria such as *Salmonella* and *Clostridium perfringens.* Probiotic supplementation also regulates the gut flora to protect the host from infections as well as to create bacteriocins, which directly prevent the growth of pathogens. This study was therefore carried out to ascertain whether the supplementation of *Pediococcus pentosaceus* GT001 can influence the growth performance, meat quality, intestinal development and cecum microbiota of *Salmonella typhimurium*-challenged broiler chickens.

**Abstract:**

This study evaluated the effects of *Pediococcus pentosaceus* GT001 on *Salmonella typhimurium* (*S. typhimurium*)-challenged broiler chickens. Two hundred Ross 708 broiler day-old chicks with comparable weight were distributed at random into four treatments with five replicates and ten chicks per replicate. The following were the treatment groups: (B) basal diet (control); (B + S) basal diet and birds were challenged with *S. typhimurium* at 1.0 × 10^7^ cfu/g; (B + P) basal diet + *Pediococcus pentosaceus* GT001 at 4.0 × 10^8^ cfu/g; (B + P + S) basal diet + *P. pentosaceus* GT001 at 4.0 × 10^8^ cfu/g and birds were challenged with *S. typhimurium* at 1.0 × 10^7^ cfu/g. There was a significant reduction (*p* < 0.05) in the body weight of the *Salmonella*-infected birds compared to the other treatment groups. However, the FCRs of the broilers were comparable among the different treatment groups (*p* > 0.05). The lipid profile and liver function indices measured were significantly enhanced in the *P. pentosaceus* GT001-supplemented groups (B + P and B + P + S) compared to the group that was *Salmonella*-challenged (*p* < 0.05) but were similar to those in the control group. The serum antioxidant activities, such as the T-AOC, SOD, CAT, GHS-Px and MDA, were significantly improved in the *P. pentosaceus* GT001-supplemented groups (B + P and B + P + S) (*p* < 0.05). The MDA was similar in the B + P and B + P + S groups, but both were significantly lower than the control and the *Salmonella* groups. The administration of *P. pentosaceus* GT001 enhanced the lipase and amylase levels in both the serum and intestine of the broilers (*p* < 0.05). The immunoglobin (IgA, IgG, IgM) and cytokine (IL-10 and IL-6) levels in the serum were significantly higher in the B, B + P and B + P + S treatment groups (*p* < 0.05). The immune-related organs (bursa and spleen) were significantly influenced in the birds fed with *P. pentosaceus* GT001. No significant variation was noted among all the dietary treatments in terms of the measured meat quality indices. The small intestinal digesta content of the *Salmonella* load was below a detectable range after 14 days of infection (*p* < 0.05). No significant differences were observed among the different treatment groups in terms of the breast pH, drip loss and meat color (*p* > 0.05). The inclusion of *P. pentosaceus* GT001 also modified the community structure in the cecum. This indicates that it has health benefits and could be incorporated in the broiler diet.

## 1. Introduction

A prevalent pathogenic bacterium that affects all animal and poultry species is *Salmonella*. It can have a negative impact on human health as well as the animal sector, particularly in the production of poultry. *Salmonella typhimurium* (*S. typhimurium*) is a common serotype that causes salmonellosis in broiler chickens. It is an intestinal bacterium that can colonize chickens [1,2]. *Salmonella* has 10 distinct serotypes, the most common of which is *Salmonella enterica* serovar enteritidis, which is found in about 60% of poultry samples [3]. Salmonellosis is a global issue for the poultry industry, especially in developing nations, and is a major cause of bacterial food poisoning in humans. It is also a key factor in lower productivity in the poultry industry [4]. A *Salmonella* infection in broiler chickens can result in lower growth rates, decreased feed efficiency and higher mortality rates. Earlier work suggests that adding multi-strain probiotics to chicken diets enhanced their intestinal microbiota and enhanced different patterns of cytokine production in response to *Salmonella* infection [5]. The increasing demand from consumers for chicken products devoid of antibiotics has increased the need to identify alternatives to antibiotics for the management of *Salmonella* infection, colonization, and carcass contamination in poultry.

The use of feed additives like probiotics and other effective management techniques, such as immunization, biosecurity protocols, and regulatory compliance, is crucial for minimizing the effect of *Salmonella* on broiler chicken production and lowering the risk of foodborne disease for consumers. Through the process of competitive exclusion, probiotics have been shown to inhibit the colonization of the gut by harmful bacteria such as *Salmonella* and *Clostridium perfringens* [6]. According to other studies, probiotics enhanced broiler chicken performance by encouraging faster growth rates and feed conversion efficiency via various mechanisms [7]. *P. pentosaceus* had a variety of probiotic benefits, the most notable of which were immunological, antioxidant, growth enhancement [8] and cholesterol-lowering properties [9]. *P. pentosaceus* significantly enhanced complement 3 expression and immunoglobin M as well as reducing the damage to the intestinal villi and goblet cells, suggesting immune system stimulation. Moreover, it possessed broad-spectrum antimicrobial qualities [8]. Several functions were enhanced by *P. pentosaceus*, including growth, immunity, diseases resistance, and the activity of digestive enzymes. In addition to regulating the gut flora to protect the host from infections, *P. pentosaceus* can create bacteriocins, which directly prevent the growth of pathogens [10]. *P. pentosaceus*, a promising strain of lactic acid bacteria (LAB), is gradually attracting attention, leading to a rapid increase in experimental research [11]. This study assessed the effect of *Pediococcus pentosaceus* GT001 on the performance, immune function, antioxidant activities and intestinal development of *Salmonella typhimurium*-challenged broiler chickens. This study was also conducted to ascertain whether *P. pentosaceus* GT001 supplementation can decrease the *S. typhimurium* load in the poultry intestine as well as affect their cecum microbiota.

## 2. Materials and Methods

### 2.1. Bacteria Strain

In an earlier investigation, the *P. pentosaceus* GT001 utilized in this experiment was isolated, cultivated and examined in vitro (as part of the Ph.D. work). After inoculating a fresh culture in MRS broth medium for an entire night at 37 °C, the cultures were centrifuged for 15 min at 3000 rpm, washed twice in sterile phosphate-buffered saline (PBS) with a pH of 7.4, and then re-suspended in PBS to bring the concentration to 4.0 × 10^8^ CFU/g. A total of 100 g of feed was well mixed with 10 mL of *P. pentosaceus* GT001.

### 2.2. Birds, Treatments, Design and Husbandry

A total of 200 Ross 708 broiler chicks at a day old and with an average weight of 39.19 g were acquired from Pluriton (Arendonk, Belgium). The chicks were split up into four experimental treatment groups at random in a completely randomized block design. The four experimental treatments comprised five replicates, with ten chicks per replicate, raised on a deep litter with an area of 3.0 m × 2.25 m. Feed and water were provided ad libitum, while the birds were maintained on a 24 h light schedule throughout the trial. A three-stage feeding regimen was implemented as follows: starter: 0–14 days of age; grower: 15–28 days of age and finisher: 29–42 days of age. The diets, comprising of maize and soybean meal in a mashed state, were formulated according to the requirements of the NRC [12] for the entire three-stage feeding regimen. Table 1 shows the nutritional values and content of the basal diets. The following were the treatment groups: basal diet (corn and soybean-based) (B); (B + S) basal diet + *S. typhimurium* challenged; (B + P) basal diet + *P. pentosaceus* − not *S. typhimurium* challenged; (B + S + P) basal diet + *P. pentosaceus* + *S typhimurium* challenged. After an hour of feed withdrawal, 100 g of basal meal was combined with about 10 mL of the probiotics, well mixed, and given to the birds (B + P and B + P + S). At the 3rd d of age, all the birds in the challenge groups (B + S and B + P + S) were orally gavaged with 1 mL of 1.0 × 10^7^
*S. typhimurium*. In accordance with conventional management procedures, the room temperature was gradually lowered to 22 °C until the completion of the experiment after being kept at 34 °C for the first five days by the use of regulated heaters, fans, and opening doors and windows. On days 14 and 21, respectively, the chicks in all the treatment groups received vaccinations against Newcastle disease (YEBIO^®^ Shandong, China) using the LaSota B1 strain of the virus (freeze-dried and live) and IBD against Gumboro disease (administered through the drinking water of the birds). The research area was regularly cleansed and de-infested to prevent infections from spreading.

### 2.3. Growth Performance

The weights of the chicks and feed were measured by replicate. The feed consumption and weight of the birds were measured weekly, and the feed conversion ratio (FCR) and body weight gain were calculated.

### 2.4. Serum Biochemistry

Using chicken-specific ELISA kits, the concentrations of total protein, albumin, globulin, creatinine, aspartate aminotransferase (AST), alanine aminotransferase (ALT), cholesterol, triglyceride, high-density lipoprotein (HDL), low-density lipoprotein (LDL), total antioxidant capacity (T-AOC), superoxide dismutase (SOD), and glutathione peroxidase (GSH-Px) activities, as well as malondialdehyde (MDA), interleukin-10 (IL-10), interleukin-6 (IL-6), and tumor necrosis factor-a (TNF-a), immunoglobulin A (IgA), immunoglobulin G (IgG), and immunoglobulin M (IgM), were measured according to the manufacturer’s protocol. The ELISA kits were purchased from the Nanjing Jiancheng Bioengineering Institute (Nanjing, China). Two birds from each replicate in each treatment were sampled for 5 mL of blood through their jugular veins, placed into vacutainer tubes and left to clot at room temperature. The serum was extracted from the clotted blood samples after centrifugation for 15 min at room temperature at 3000 rpm.

### 2.5. Intestinal Measurement and pH Assessment

Using a measuring tape, the duodenum, ileal, and jejunal lengths were measured in centimeters. The intestinal contents of the duodenum, ileum, and jejunum obtained from the slain birds (two bird per replicate) at day 42 were put into sterile plastic containers. A pH probe was then inserted directly into the digesta content to record the pH (SP-701/pH/mV/Temp.Meter, Suntex, Taipei, Taiwan).

### 2.6. Digestive Enzyme Measurement

With ELISA kits obtained from the Nanjing Jiancheng Bioengineering Institute (Nanjing, China), the serum digestive lipase and amylase enzymes were measured according to the manufacturer’s instructions. The intestinal digestive enzymes of the small intestinal digesta was homogenized using 0.9% physiological saline. The homogenate was then centrifuged at 5000× *g* for 15 min to collect the supernatant. Subsequently, the Nanjing Jiancheng detection kits were used to measure the lipase and amylase activity in the small intestinal digesta.

### 2.7. Intestinal Morphology Measurement

At day 42, portions from the middle of the duodenum, jejunum, and ileum were taken from the birds. These segments were then washed with a 0.9% salt solution and preserved for 48 h in a 10% formaldehyde–phosphate buffer. The slices were then stained with hematoxylin–eosin to allow for the measurements of the height and width of the intestinal villi as well as the depth of the intestinal crypts under a light microscope. Ten full, precisely aligned crypt–villus units were selected in triplicate for each intestinal cross-section and used to calculate the crypt depth to villus height ratio. The histological segments were examined using a Leica DM500 light (Wetzlar, Germany). The crypt depth was measured from the root of the lower limit of the crypt to the villi–crypt junction, while the villi height was measured vertically from the villi–crypt junction to the tip of the villi [13].

### 2.8. Organs and Meat Quality Assessment

At day 42, 2 birds similar to the mean weight were selected from each replicate for the measurement of the organs and meat quality. Following the process of euthanasia, the jugular vein was cut off, feathers and head shanks were removed and the remainder of the carcasses was dissected. For the purpose of measuring the meat quality, each bird’s left and right breasts were used. Using a microprocessor pH-meter (SevenCompact pH Meter, S220, Mettler Toledo, Columbus, OH, USA), the breast muscle’s pH was determined 15 min after slaughter. The starting and final weights of each sample were utilized to quantify the drip loss of the chicken breast meat samples and subsequently suspended and standardized for the surface area in cups at 4 °C for 48 h [14]. Using a Minolta colorimeter, the samples’ meat colors were measured at three distinct points across the breast flesh and reported as lightness (L*), redness (a*), and yellowness (b*) [14].

### 2.9. Salmonella and TVC Enumeration

Samples of the small intestinal digesta were aseptically collected, placed in sterile plastic containers and covered with liquid nitrogen for laboratory assessment. The digesta samples were stored at −40 °C until the microbial count examination. One gram of the digesta samples from the small intestine was serially diluted ten times in the lab using nine milliliters of peptone water. The samples that had been diluted (0.1 mL) were added to selective agar and the bacterial enumeration was determined in a biosafety cabinet. *Salmonella* was incubated using XLD agar, while TVC was incubated using Standard Plate Count agar. The microbial population was represented as log10 colony-forming units/g of the digesta.

### 2.10. Analysis of Cecum Microbial Ecology

On day 42, 5 cecum content samples were taken from each treatment, beaded with a Mini-BeadBeater for DNA extraction, and the DNA was extracted using the Power Fecal DNA Isolation Kit (MO BIO, Carlsbad, CA, USA). The DNA was quantified using a NanoDrop spectrophotometer (Nyxor Biotech, Paris, France) and stained using the Quant-iT Pico Green dsDNA Kit (Invitrogen Ltd., Paisley, UK). In order to perform the DNA MiSeq sequencing, the V4 region of the bacterial 16S rDNA was amplified by PCR using the universal primers 515F (50-CCTACGGGNGGCWGCAG-30) and 806R (50-GGACTACHVGGGTWTCTAAT-30), which included the sequence of a sample bar and the FLX Titanium adapters. The cycling parameters were as follows: 4 min of initial denaturation at 94 °C, 25 cycles of denaturation at 94 °C (30 s each cycle), 45 s of annealing at 50 °C, 30 s of elongation at 72 °C, and 5 min of final extension at 72 °C. For the MiSeq sequencing, three distinct PCR reactions were combined for each sample. The PCR products were isolated using 1.5% agarose gel electrophoresis and further purified with QIAqu. The PCR products were purified using a Gel Extraction Kit (Qiagen, Hilden, Germany). The Quant-iT Pico Green dsDNA Assay Kit (Invitrogen, Carlsbad, CA, USA) was used to quantify the amplicons. Each sample’s amplicons were combined at the same concentrations. The MiSeq Reagent Kit v2 (Illumina, San Diego, CA, USA) was used for the MiSeq sequencing, and libraries were produced using the TruSeq DNA PCR-Free Sample Prep Kit (Ilumina, San Diego, CA, USA). Paired-end readings were assigned to the samples with unique barcodes; the barcode and primer sequence were snipped. FLASH (https://ccb.jhu.edu/software/FLASH/MANUAL, accessed on 1 June 2024) was used for merging the paired-end readings. The length and quality of the resulting sequences were further examined and filtered. To obtain clean, high-quality tags in line with fqtrim (v0.94), the raw reads were put through quality filtering under predetermined filtering settings. The program V-search (v2.3.4) was utilized to filter the chimeric sequences. Using DADA2, (https://benjjneb.github.io/dada2/, accessed on 1 June 2024) the feature table and feature sequence were obtained after dereliction. Using the feature abundance, the relative abundance of each sample was normalized in line with the SILVA (version 132) classifier. The graphs were created using the R program, and the analysis of the alpha and beta diversities was computed using QIIME2 (https://view.qiime2.org/, accessed on 1 June 2024).

### 2.11. Statistical Analysis

Using a randomized complete block design (RCBD) with four replications, the general linear model (GLM) of Minitab^®^ version 18.1 (Minitab version 18) was used to analyze the data as a one-way ANOVA. Tukey’s test was used to determine whether the treatment means differed from one another, and statistical significance was defined as *p* < 0.05. All the data are presented as the mean ± SEM. To reveal the similarities and contrasts between the four treatment groups on OTUs, a Venn diagram was used. To obtain the comparative study of the intergroup and group differences in terms of the individual fraction distance, the beta diversity analyses for the principal coordinate analysis (PCoA), the principal component analysis (PCA), and the nonmetric multidimensional scaling (NMDS) were employed. The permutational analysis of variance (PERMANOVA) was used to examine the beta diversity. The relative abundances at the phylum, family, and genus were examined using the Kruskal-Wallis test (*p* < 0.05).

## 3. Results

### 3.1. Growth Performance

The production parameter outcomes are presented in Table 2. The birds’ initial weights prior to the trial started were comparable for all the feeding regimens. There was no significant influence of *P. pentosaceus* GT001 supplementation on the daily feed intake (*p* = 0.677) and feed conversion ratio (*p* = 0.252) of the birds. The *Salmonella*-infected birds (B + S) showed significant (*p* < 0.05) decreased body weight changes (final weight, total weight gain and average daily gain) during the study. The body weight changes of the B, B + P and B + P + S groups were similar (*p* > 0.05) but significantly higher (*p* < 0.05) compared to the B + S group. The *Salmonella*-infected groups of B + S and B + P + S recorded mortality rates of 6.7% and 3.3%, respectively, compared to both the B- and B + P-treated birds.

### 3.2. Serum Biochemistry

The results of the liver function and lipid profile are presented in Table 3. Significant differences (*p* < 0.05) were observed among all the indices measured. The B and B + P birds had similar albumin and creatinine content but were higher (*p* < 0.001) compared to their counterpart in the B + S and B + P + S groups. The globulin level of *P. pentosaceus* GT001 supplementation (B + P) was higher (*p* = 0.004) than B + S and B + S; nonetheless, it was similar to the B. *Salmonella*-infected group (B + S), which had the least significant total protein content, while the *P. pentosaceus* GT001 supplementation group (B + P) recorded the highest significant protein content (*p* = 0.001). The use of B + P in terms of the protein content was similar to B, while B + S showed similarity with the B + P + S-treated birds. With regards to the ALT and AST content, *P. pentosaceus* GT001 supplementation (B + P) recorded higher levels, while B + S recorded the lowest (*p* < 0.001) In terms of the T cholesterol, TG and HDL, the *P. pentosaceus* GT001-supplemented groups (B + P and B + P + S) produced similar results, though lower (*p* < 0.001) compared to the groups of B and B + S. The LDL content for both the B + P and B + P + S groups were similar, though both were higher (*p* < 0.001) than the B and B + S groups.

### 3.3. Serum Antioxidant Activities

Table 4 shows the outcome of the serum antioxidant activities. There was a significant variation among all the treatment groups for all the indices measured under serum antioxidant activities. The birds treated with B + P recorded the highest significant (*p* = 0.001) T-AOC level compared to the *Salmonella*-infected groups. Furthermore, the B group showed similarity with the *Salmonella*-infected groups. The birds on the B + P produced the highest and most significant (*p* < 0.001) SOD and GHS-Px compared to the other treatments, while the use of B + S produced the least level. The MDA contents of the *P. pentosaceus* GT001-supplemented groups (B + P and B + P + S) were comparable but (*p* < 0.001) lower than both group of B and B + S. Additionally, B + P was higher (*p* < 0.001) than B + P + S, B and B + S in terms of the CAT content, while the B and B + S groups showed comparable levels.

### 3.4. Serum Cytokines and Immunology

There were significant variations among the treatment groups in terms of the serum cytokines and immunology parameters, as shown in Table 5. The values obtained for TNF-α and IL 6 were higher in B + P, with B + S having the least significant levels (*p* < 0.01). The use of B + P produced the highest significant (*p* < 0.01) IL 10 content compared to B, B + S and B + P + S. However, the choice of the B, B + S and B + P + S treatments showed comparable levels during the study. No significant differences were noted in the serum IgG and IgA of treatments B, B + P and B + P + S, though a high significant variation was observed when compared to BD + S (*p* < 0.05). Regarding the serum IgM, B and B + P recorded the highest significant values, followed by B + P + S, with B + S recording the least value (*p* < 0.01).

### 3.5. Digestive Enzymes

Table 6 presents the results of the serum and intestinal digestive enzymes of the broiler chickens. In terms of the serum amylase content, the *P. pentosaceus* GT001-supplemented groups (B + P and B + P + S) were similar but (*p* < 0.01) higher compared to both the BD and BD + S groups. However, the birds in the B + S group recorded the least. Similarly, the serum lipase content was higher (*p* < 0.01) in the *P. pentosaceus* GT001-supplemented groups (B + P and B + P + S). A significant difference was noted in intestinal amylase. B + P had the highest significant value, followed by B, while the *Salmonella*-infected groups (B + S and B + P + S) recorded the least value (*p* < 0.05). With intestinal lipase, the *P. pentosaceus* GT001-supplemented groups (B + P and B + P + S) had the highest significant values, but B + P + S was similar to B and B + S had the least significant value (*p* < 0.01).

### 3.6. Organs

The results of the internal organ are shown in Table 7. No significant differences (*p* > 0.05) were observed among the dietary treatments in the measured organ parameters expect the immune-related organs (spleen, bursa) and small intestine. Regarding the bursa, the *P. pentosaceus* GT001-supplemented groups (B + P and B + P + S) and the B group showed similarity but were higher than the *Salmonella*-infected group (B + S) (*p* = 0.009). B + P and B were similar in terms of the spleen, though B + P was significantly higher compared to the *Salmonella*-infected groups (B + S and B + P + S). The B group was similar to B + P + S but higher than that of the B + S group. The small intestinal weights of B, B + P and B + P + S were high (*p* = 0.043), while B + S recorded the least, though similar to B.

### 3.7. Meat Quality

The results in Table 8 show the impact of *P. pentosaceus* GT001 on meat quality. No significant variation was noted among all the dietary treatments in terms of the meat quality indices measured (*p* > 0.05).

### 3.8. Intestinal pH and Length

The outcomes concerning the intestinal length and pH are presented in Table 9. The intestinal length showed no significant differences among the treatments (*p* > 0.05). The lengthiest duodenum was observed in B + P, followed by B, then B + P + S, with B + S having the least. With the jejunal length, the B group recorded the highest value during the period of the trial, while B + S recorded the lowest length. Similarly, the longest ileum was noted in the B group, while the B + S group recorded the least. However, significant variations were noted in the intestinal pH. Both the B + P and B treatments recorded the highest duodenal pH, which were similar but higher than the B + S group (*p* < 0.01). The pH of the jejunum of B was higher compared to the other treatments (*p* < 0.01). The *Salmonella*-infected and *P. pentosaceus* GT001-supplemented groups (B + S, B + P and B + P + S) showed no significant difference during the study. A significance difference was noted in the ileum pH. The B groups showed the highest significant pH value compared to the other treatments (*p* < 0.01). The pH of B + S was also higher than the *P. pentosaceus* GT001-supplemented groups (B + P and B + P + S) (*p* < 0.01).

### 3.9. Small Intestinal Morphology

The results of the morphology of the broilers’ small intestines are shown in Table 10. The duodenal VH and CD varied significantly among the treatments, while a significant difference was not noted in the VH:CD. Treatment B was higher compared to the other treatments, followed by the *P. pentosaceus* GT001-supplemented groups (B + P and B + P + S), while the *Salmonella*-infected group (B + S) produced the least significant value (*p* < 0.01). However, the duodenal CD values were comparable among the treatments of B, B + P and B + P + S, though all were higher than the B + S group (*p* = 0.001). The jejunal VH of B recorded the highest significant difference, followed by B + P and B + P + S, while B + S recorded the least significant difference (*p* < 0.001). With the CD of the jejunum, the highest significant difference was noted in B + P, while the lowest significant difference was observed in the B + S group. The VH:CD levels were comparable among the dietary treatments. Significant differences were noted among the dietary groups in the ilea VH and CD values, while no significant difference was observed in the VH:CD value. Treatment B + P was higher (*p* < 0.001) in terms of the ilea VH and CD when compared to the other treatments. The VH and CD of B + P + S and B were similar, but both were higher than that of B + S.

### 3.10. Salmonella Load

The outcomes of the *Salmonella* count and total viable count during different periods are presented in Table 11. Significant variations were noted in both parameters at the different periods post infection. B + S, which was the *Salmonella*-infected group, recorded the highest significant *Salmonella* level compared to the other groups (*p* < 0.01) at day 3, 7 and 14 post infection. No significant variation (*p* > 0.05) was observed among B and the *P. pentosaceus* GT001-supplemented groups (B + P and B + P + S). In terms of the total viable count, the *Salmonella*-challenged + probiotic (B + P + S) group recorded the highest significant value at day 3 post infection. The TVC at day 3 post infection of the B + P and B + S groups were comparable, but both were higher than the B treatment group (*p* < 0.01). The highest significant TVC value at day 7 post infection was noted in the B and B + S treatment groups, while the probiotic-supplemented groups (B + P and B + P + S) recorded the least values. At day 14 post infection, B observed the highest significant value of TVC, followed by B + S, while B + P and B + P + S recorded the least TVC values.

### 3.11. Analysis of Cecum Microbial Ecology

#### 3.11.1. Microbial Effective Sequence

Figure 1 displays a Venn diagram based on the OTUs in the total sequences of each group. The Venn diagram explains the overlap (639 OTU, core) with the four displayed groups combined. Across all four treatments, a total of 435, 310, 530 and 430 unique OTU were exposed. To be precise, 60 OTU were found in both the B and B + S treatments; 49 OUT were found in both the B + S and B + P treatments; 49 OUT were found in the B + P and B + P + S treatments, and 60 OTU were found in the B + P + S and B treatments

#### 3.11.2. Microbial Diversity

Table 12 displays the indicators of the alpha diversity of the following indices: Goods coverage, Shannon, Simpson, Chao1 and Observed OUTs. No significant differences were observed in all the indices measured among the dietary treatments (*p* > 0.05). Figure 2 displays the beta diversity indicators of the PCoA, PCA, and NMDS that were obtained to calculate the intragroup and intergroup distances. It was observed that the differences among the treatments were not significant in the PCA (Figure 2A), weighted PCoA (Figure 2B), and unweighted NMDS (Figure 2E) plots. However, a significant difference was estimated in the unweighted PCoA (Figure 2C) and weighted NMDS (Figure 2D) plots (*p* < 0.05). The statistically significant *p* values were estimated in the intragroup and intergroup distances.

#### 3.11.3. Microbial Composition

The effects of the dietary supplementation of *P. pentosaceus* GT001 on the phylum, family and genus taxa levels in the microbial content of the cecum are presented in Figure 3A–C. The estimation of the phylum level is shown in Figure 3A, where *Proteobacteria*, *Bacteroidetes* and *Firmicutes* were the most dominant phyla of the cecal community, accounting for about 3.32%, 19.06% and 75.61%, respectively. *Rikenellaceae*, *Lachnospiraceae* and *Ruminococcaceae* were the dominant families in the cecal community at the family level, as presented in Figure 3B. *Rikenellaceae* accounted for 8.02%, *Lachnospiraceae* recorded 10.05% and *Ruminococcaceae* accounted for the highest, which was 43.93%. In terms of the genus taxonomy, *Ruminococcaceae_UCG-005*, *Ruminococcaceae_UCG-014*, *Alistipes* and *Faecalibacterium* accounted for around 5.60%, 7.43%, 8.02%, and 12.12% of the genera in the cecal community, respectively, as presented in Figure 3C.

Some significant differences were observed in the relative abundance at the phylum, family and genus levels (Table 13). The abundance of the *Cyanobacteria* phylum was similar between the B and B + P treatments, but both were higher compared to the *Salmonella*-challenged groups (B + S and B + P + S) (*p* = 0.033). The B + P + S treatment recorded the highest significance for the *Actinobacteria* phylum, while the other treatments were similar (*p* = 0.002). *Proteobacteria* at the phylum level was similar among the dietary treatments (*p* = 0.615). Significance was noted among the treatments in terms of *Clostridiaceae* and *Lactobacillaceae* at the family level. *Clostridiaceae* was higher in the probiotic-supplemented group (B + P), but similarities were observed among B, B + S and B + P + S (*p* = 0.054). No significant differences were noted among the treatments with regard to *Peptostreptococcaceae* and *Bacteroidaceae* at the family level. At the genus level, the control and probiotic-supplemented groups (B and B + P) recorded the highest significant values in terms of *Lactobacillus* and *GCA-900066575* compared to the *Salmonella*-challenged groups (B + S and B + P + S) (*p* < 0.05). Also, the control treatment (B) was significantly higher compared to B + S, B + P and B + P + S in regards to *Escherichia-Shigella* and *CHKCI001* genus. *Clostridiales_unclassified*, *Anaerofustis*, *CHKCI002*, *Papillibacter* and *GCA-900066225* genus were similar among the dietary treatments (*p* > 0.05).

## 4. Discussion

### 4.1. Growth Performance

The use of probiotics as a feed additive is widely accepted as a preventive measure in reducing pathogen infection and enhancing the growth performance of poultry [15]. To assess the effectiveness of *P. pentosaceus* GT001 on the growth performance, this study noted the decreased body weight of the *Salmonella*-infected birds. *Salmonella* infection resulting from the damage of the internal mucosal affects feed absorption and reduces intestinal motility [16,17]. The positive effects of *P. pentosaceus* GT001 as a probiotic on the body weight changes of broilers are in agreement with previous studies by Sikandar et al. [18] and Chang et al. [19]. In the present study, the FCR was similar between the *Salmonella*-infected birds and the probiotic-supplemented birds. This is in agreement with the findings of Mountzouris et al. [20], who reported similarity in terms of the FCR of *Salmonella*-infected and probiotic birds. On the contrary, previous studies have reported superior FCRs in probiotic supplemented birds [18,19]. These inconsistent outcomes may potentially stem from variations in the type, quantity, or dosage of *Salmonella* delivered, which resulted in varying degrees of intestinal environment stability [21]. Additionally, the high mortality in the *Salmonella*-challenged birds in the present study is attributable to the *Salmonella* infection at day 3.

### 4.2. Serum Biochemistry Activities

There was an enhancement in all the lipid profile and liver function indices measured. The probiotic *P. pentosaceus* GT001-supplemented groups had better values than the *Salmonella*-challenged group. Our results were in accordance with the results of the study performed by de Azevedo et al. [22], which showed that *P. pentosaceus* stimulates the immune system, decreases the HDL, total cholesterol and TG levels, increases LDL and improves the digestion of protein. The purpose of HDL is to convey any residual cholesterol that is not being used to the liver. The leftover cholesterol will be used as a constituent in the production of bile salt and steroid hormones, while the remaining inactive cholesterol will be defecated. Alterations in the activities of AST and ALT are also specific indicators that can be utilized to ascertain the organism’s hepatocyte activity as well as specific indicators of hepatocyte damage [23]. In the current study, the AST and ALT activity levels obtained among the treatment groups were within the normal range, though the probiotic-supplemented groups were significantly high. The range of AST and ALT activities among the treatments during the study indicates that feeding *P. pentosaceus* GT001 to broilers resulted in normal liver function.

### 4.3. Serum Antioxidant Activities

The *P. pentosaceus* GT001-supplemented groups showed significant responses in all the antioxidant activity indices measured. There was an increase in the serum T-AOC, SOD, CAT and GHS-Px activities, which is similar to the observation in the research by Mohamed et al. [24] and Zhang et al. [25] when probiotic was fed to broiler chickens. The serum MDA activity was decreased in the *P. pentosaceus* GT001-supplemented group in the current the study. Mohamed et al. [24] and Wang et al. [26] previously reported a reduced MDA in probiotic-fed birds. This observation may arise from the enhanced serum SOD activities of the birds. It is known that alterations in the activity of a number of antioxidant enzymes can be employed to evaluate an animal’s overall antioxidant state and degree of oxidative stress. According to Yang et al. [27], the T-AOC reflects the organism’s level of antioxidants and SOD catalyzes the conversion of the superoxide anion into hydrogen peroxide. Contrary to the current study, Abudabos et al. [28] and Erdoğan et al. [29] reported comparable values among treatments, indicating that either probiotics or *Salmonella* had no effect on the overall antioxidant capacity or oxidative stress. Factors including the type of challenge, dosage and the probiotic strain could account for this observed variation.

### 4.4. Cytokines and Immunological Activities in the Serum

Immunological indices serve as the first line of defense against invasive pathogens since many microbial pathogens first come into touch with their hosts via mucosal surfaces, particularly in the alimentary canal [30]. Therefore, regular and moderate use of probiotics can have a significant influence on the immune system and boost the immunoglobulin concentration. The immunoglobulin (IgG, IgA and IgM) levels in the current study were improved in the probiotic group and probiotic + challenged group. The *P. pentosaceus* augmented endogenous interferon and cytokine production, boosting humoral as well as cell-mediated immunity [31,32]. A similar observation was noted in the serum cytokines indices. The increase levels of the anti-inflammatory cytokines IL-10 and IL-6 may be due to the suppression of the stress-related inflammatory response when the probiotic was the supplement in the diet [33]. In accordance with the present study, Masuda et al. [34] observed highly induced cytokines when the *P. pentosaceus* strain was supplemented in a diet.

### 4.5. Digestive Enzymes

The administration of probiotic *P. pentosaceus* GT001 enhanced the lipase and amylase activities in both the serum and intestine in this study, in accordance with the previous studies by Mohamed et al. [35] and Wang and Gu [36]. The authors noted the improved amylase and lipase activities of the broiler diet supplemented with probiotics. The digestion of protein, carbohydrate, and fats is improved by higher lipase and amylase activity, which may have contributed to the improvement in growth observed in this study.

### 4.6. Organs and Meat Quality

A significant rise was noted in the weights of the bursa of Fabricius and spleen, unlike the other organs measured, in which no differences in weight were observed in the birds fed with *P. pentosaceus* GT001. Given that T and B lymphocyte maturation sites are located in the bursa of Fabricius in birds, the size and mass of this organ can reveal crucial general information about the maturation and development of the immune system [37]. According to Mohamed et al. [35] and Park and Kim [37], birds fed a meal supplemented with probiotics showed more weight in the bursa of Fabricius than the control groups. As a result, from our study, the apparent expanded bursa of Fabricius in the groups administered with *P. pentosaceus* GT001 may be a favorable signal for the immune system development of the birds. Thus, the apparent larger bursa of Fabricius in the *P. pentosaceus* GT001-fed groups in our study may have a positive impact on the immune system development of the birds at the start of the study.

Probiotics have been used to improve meat quality but have shown inconsistent results [38,39,40]. No significant differences were noted among the treatment groups in terms of the breast pH and drip loss measured after 48 h post slaughter. Drip loss is a significant variable in the assessment of meat quality because water loss can result in the loss of some nutrients in the fluid, which can affect the meat’s softness, flavor, and juiciness [41]. When choosing poultry meat, consumers consider color to be one of the most important quality factors.

Numerous factors, like diet and genetics, affect the color of meat [42]. However, the present study did not observe any influence of *P. pentosaceus* GT001 on meat color. Contrary to our current study, Park and Kim [37] and Macelline et al. [38] reported enhanced meat quality when the diet of broiler chickens was supplemented with probiotics. These inconsistent results could be due to variations in the type and dosage of probiotic used [21].

### 4.7. Intestinal Length and pH

Healthier small intestinal pH values were observed in the *P. pentosaceus* GT001-fed group, while a similar length of small intestine was noted among all the treatment groups. Numerous theories have been proposed to explain the beneficial effects of probiotics in reducing the pH of the gut, which leads to a decrease in the stability of the dangerous bacteria in the intestines [43]. Probiotics can lower the pH of the digestive system and disrupt the ideal pH range 7 for *Salmonella* environments [43]. As a result, birds’ performance, feed conversion, and growth rate can all be improved. This finding is in accordance with that reported by Chen et al. [44], who reported that *P. pentosaceus* supplementation significantly decreased the pH value in the gut.

### 4.8. Intestinal Morphology

The birds fed *P. pentosaceus* GT001 had longer duodenal, jejunal and ilea villi and crypts compared to the *Salmonella*-infected birds in the present study. Therefore, the decline in production performance in the birds challenged with *Salmonella* can be explained by decreased villus height since a longer villus is a sign of a healthy gut. Research [45] shows that shorter villi and deeper crypts result in less disease resistance, increased mucus secretion in the GIT, poor nutrient absorption, the presence of toxins, and generally poorer broiler performance. Shalaei et al. [46] reported that birds fed a diet containing probiotic had higher villi height in the duodenum than birds fed other additives.

### 4.9. Salmonella Load and Total Viable Count

The small intestinal digesta content of the *Salmonella* load was below a detectable range after 14 days infection. The *Salmonella* load in the birds in the probiotic + challenge group was similar to that of the probiotic and control. Probiotics have been shown by researchers [20,37] to competitively exclude pathogens in the chicken gut. According to the competitive exclusion theory of Patterson and Burkholder [47], beneficial gut bacteria compete with pathogenic bacteria for receptor sites and nutrition before producing antimicrobial chemicals such as bacteriocins to reduce their burden in the intestine. Probiotics also enhance the function of the mucosal barrier, which reduces the load of *Salmonella* in the gut [48]. According to these findings, *P. pentosaceus* GT001 may be used as a probiotic to treat *Salmonella* infections in poultry. Consistent with the current study, several researchers [19,49] also reported a reduced *Salmonella* load in the GIT when the broiler diet was supplemented with probiotics. Ultimately, the findings demonstrated that probiotic *P. pentosaceus* GT001 effectively improved the microbiota and *Salmonella* load of birds that were exposed to *Salmonella*.

### 4.10. Cecum Microbial Ecology

Dietary inclusion of *P. Pentosaceus* GT001 in this study changed the broiler microbiota and enhanced the immunity against *Salmonella* infection. When Zhang et al. [50] supplemented broiler chicks’ diets with probiotics, they observed no discernible changes in the microbiota’s alpha diversity (Shannon, Simpson and Chao1) at 42 days of age in the cecum. The microbiome community’s diversity is described by the Simpson and Shannon indices, but the richness of the microbial diversity is reflected by the Chao1 index and observed species [51].

At the phylum level, the majority of the broiler microbiota was made up of *Firmicutes* and *Bacteroidetes*, accounting for about 95% of the total cecum microbiota. These organisms are involved in energy production and metabolism, specifically the fermentation of microbes and the digestion of starch [52]. Also, in the broilers infected with *Salmonella*, the greater presence of *Proteobacteria* points to gastrointestinal dysbiosis and imbalance, but the abundance of *Proteobacteria* was similar among the treatments. The relative abundance of *Cyanobacteria* phyla was increased in the B and B + P groups. Contrary to our study, Mohamed et al. [24] and Trela et al. [53] reported a decrease in *Cyanobacteria* in the microbial content of the cecum when birds were fed probiotics. Some organisms within the phylum *Cyanobacteria* are capable of producing neurotoxins that can lead to diseases and death [54]. This study observed that the level of the families of *Rikenellaceae*, *Lachnospiraceae* and *Ruminococcaceae* dominated the cecum, amounting for a total of 62%. Because the families *Ruminococcaceae*, *Rikenellaceae* and *Lachnospiraceae* are associated with the production of short-chain fatty acids, they enhance the feed conversion ratio [55]. This could explain why the feed conversion was similar across the dietary treatments, leading to the enhanced growth performance of the *Salmonella*-challenged bird plus probiotic supplementation. Zhou et al. [56] also reported that these three families are the most abundant in the small intestine, which is in consistent with our findings.

In terms of the cecum community, the key bacteria genera observed in this study were *Ruminococcaceae_UCG-005*, *Ruminococcaceae_UCG-014*, *Alistipes* and *Faecalibacterium* and they constituted about 33.17%. In accordance with research by Mohamed et al. [24] and Zhu et al. [57], *Ruminococcaceae_UCG-014*, *Alistipes* and *Faecalibacterium* are the most abundant genera of bacteria in the microbial community of the cecum. These genera are members of the *Firmicutes* family, which has been associated with enhanced weight gain in chickens and an increase in the rate of nutrient absorption by creating compounds that the gut wall can absorb as an energy source [58]. This assertion can be associated with the enhanced performance in the *Salmonella*-challenged birds + probiotic supplementation treatment. The relative abundance of the genera *Escherichia-Shigella* was inhibited by *Pediococcus pentosaceus* GT00l in the cecal community, while *Clostridiales_unclassified* was similar among the treatments. In accordance with this study, Mohamed et al. [24] observed a dramatic decrease in the relative abundance of genera of the *Escherichia–Shigella* when the broiler diet was supplemented with either probiotic or antibiotic. According to Ma et al. [59], *Escherichia–Shigella* is an opportunistic pathogenic bacterium that can cause intestinal destruction and measure pro-inflammatory activities through a variety of means, including the spread of virulence factors. This increases the host’s risk of infection. In the gut, *Clostridium_unclassified* can cooperate and compete with other bacteria to proliferate. According to Bertoluzzi et al. [60], *Clostridia_unclassified* generates some toxins that might lead to severe illnesses in poultry. Bacteriocins, which *Pediococcus pentosaceus* is able to produce, have been reported to exhibit antibacterial activity in a number of models [61]. *Pediococcus pentosaceus* is a lactic acid bacterium that also has the ability to create organic acid. According to Cui et al. [62], lactic acid can prevent pathogenic bacteria from colonizing the GIT. In the B and B + P groups, the relative abundance of the *Lactobacillus* genus was increased. One of the most important bacteria probiotics is *Lactobacillus* due to it suppression of pathogens, promotion of the growth of beneficial bacteria and enhancement of growth performance. It also assists in the maintenance of the microbial balance in the intestine [63].

## 5. Conclusions

In conclusion, this study demonstrates *P. pentosaceus* GT001 to be a probiotic and feed additive that has health benefits and could be incorporated in the broiler diet. Dietary inclusion of *P. pentosaceus* GT001 as a probiotic led to significant changes in the growth performance, immune function, digestive enzymes and intestinal morphology of *Salmonella typhimurium*-challenged broiler chickens. Additionally, the inclusion of *P. pentosaceus* GT001 modulated the community structure in the cecum and reduced the *Salmonella* load in the small intestine of the *Salmonella typhimurium*-challenged birds. The inclusion of *P. pentosaceus* GT001 in the poultry diet has the potential to reduce the *Salmonella* load and produce healthier broiler birds.

## Figures and Tables

**Figure 1 animals-14-01676-f001:**
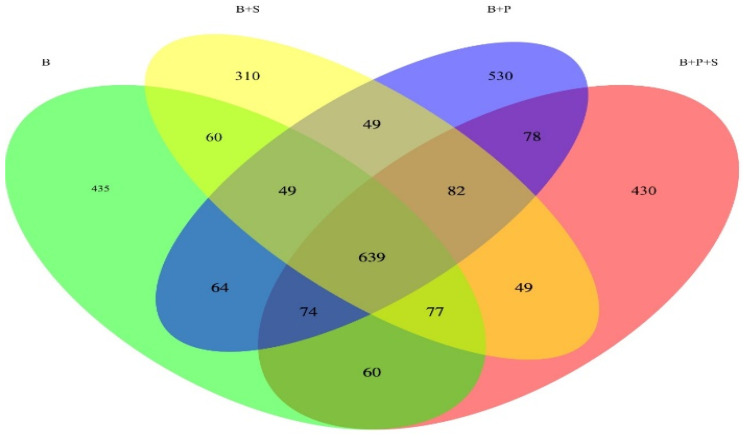
Venn diagram based on the OTUs in the total sequences of each treatment. B—basal diet (corn and soybean-based); B + S—basal diet + *Salmonella typhimurium* challenged; B + P—basal diet + *P. pentosaceus* − not *Salmonella typhimurium* challenged; B + P + S—basal diet + *P. pentosaceus* + *Salmonella typhimurium* challenged.

**Figure 2 animals-14-01676-f002:**
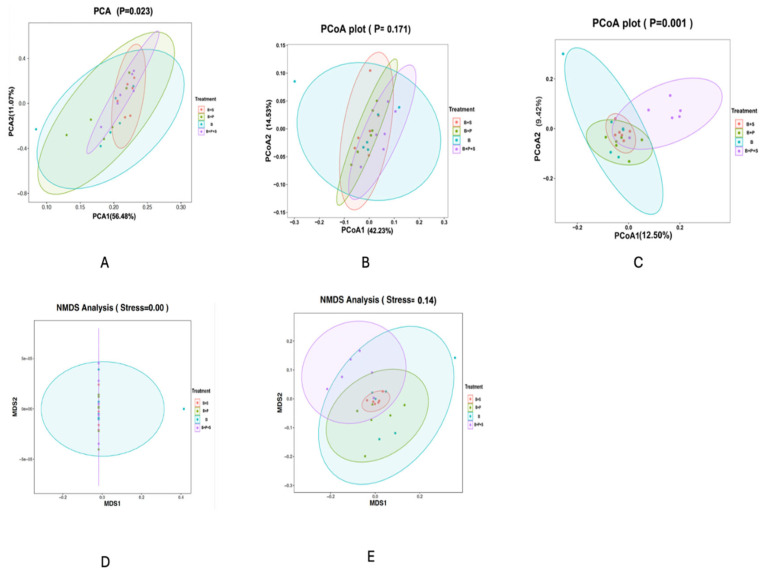
Effect of dietary inclusion of *P. pentosaceus* GT001 on the cecal microbiota composition of broilers by indices of beta diversity. (**A**) PCA; (**B**) weighted PCoA; (**C**) unweighted PCoA; (**D**) weighted NMDS; (**E**) unweighted NMDS. B—basal diet (corn and soybean-based); B + S—basal diet + *Salmonella typhimurium* challenged; B + P—basal diet + *P. pentosaceus* − not *Salmonella typhimurium* challenged; B + P + S—basal diet + *P. pentosaceus* + *Salmonella typhimurium* challenged.

**Figure 3 animals-14-01676-f003:**
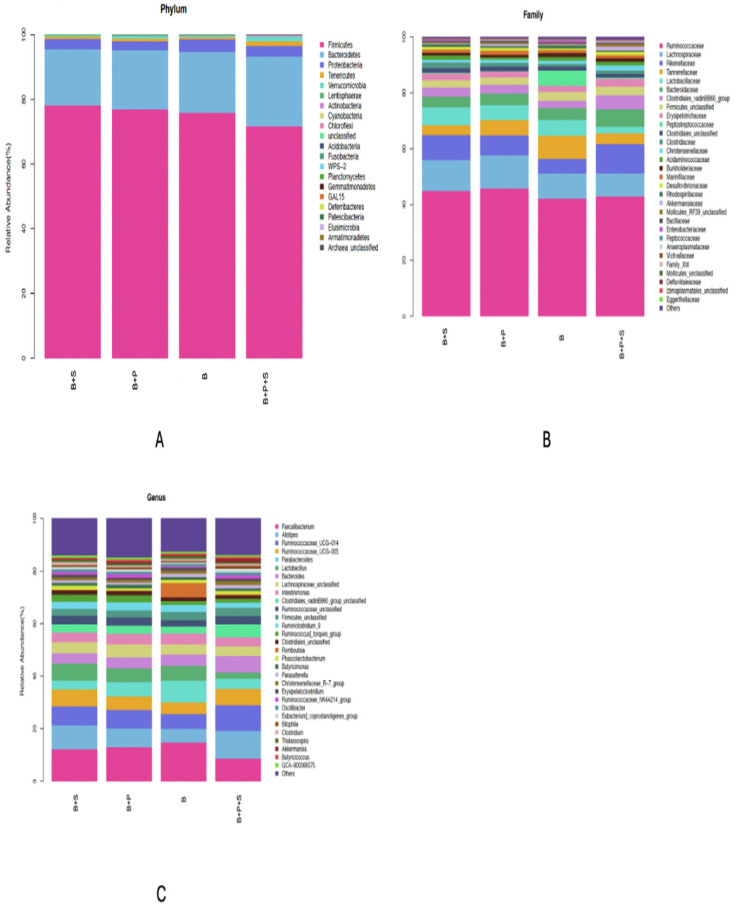
Effect of dietary inclusion of *P. pentosaceus* GT001 on the relative abundance percentage of cecal microbiota communities at the (**A**) phylum, (**B**) family and (**C**) genus levels. B—basal diet (corn and soybean-based); B + S—basal diet + *Salmonella typhimurium* challenged; B + P—basal diet + *P. pentosaceus* − not *Salmonella typhimurium* challenged; B + P + S—basal diet + *P. pentosaceus* + *Salmonella typhimurium* challenged.

**Table 1 animals-14-01676-t001:** Ingredients and chemical composition of the experimental diets.

Items (kg)	Starter (0–14 Days)	Grower (15–28 Days)	Finisher (29–42 Days)
Maize	59.2	62.0	64.0
Soybean meal (48% CP) ^a^	30.0	27.5	25.5
Vegetable oil	2.0	2.0	2.0
Premix ^b^	1.0	1.0	1.0
DL-methionine	0.2	0.2	0.2
L-lysine	0.2	0.2	0.2
Salt	0.3	0.3	0.3
Dicalcium phosphate	0.5	0.5	0.5
Fish	5.3	5.0	5.0
Limestone	1.3	1.3	1.3
Total	100	100	100
Calculated nutrient value			
Energy (Kcal/kg) ^c^	3285	3286	3288
Crude protein (%)	22.19	21.09	20.34
Total phosphorus (%)	0.73	0.69	0.65
Methionine (%)	0.24	0.23	0.23
Methionine + Cysteine (%)	0.93	0.90	0.89
Lysine (%)	1.40	1.31	1.26
Ether extract (%)	5.25	5.29	5.33
Crude fiber (%)	3.06	2.97	2.89

^a^ Crude protein ^b^ Premix: provided the following per kilogram of diet: 13,000 IU of vitamin A; 1300 IU of vitamin D; 65 IU of vitamin E; 3.4 mg of menadione; 37 mg of pantothenic acid; 6.6 mg of riboflavin; 3.7 mg of folic acid; 39 mg of niacin; 1.0 mg of thiamine; 4.3 mg of vitamin B6; 0.23 mg biotin; 0.075 mg of vitamin B12: 43 mg of choline chloride. 170 mg of zinc; 140 mg of iron; 34 mg of manganese; 16 mg of copper; 0.29 mg of iodine; 0.29 mg of selenium. ^c^ Metabolizable energy.

**Table 2 animals-14-01676-t002:** Effect of *Pediococcus pentosaceus* GT001 on the production performance of broilers.

Items	Treatments ^1^				SEM ^2^	*p*-Value
	B	B + S	B + P	B + P + S		
Initial weight (g)	39.3	39.4	38.8	39.4		
Final weight (g)	2465.4 ^a^	2254.6 ^b^	2477.7 ^a^	2385.0 ^a^	36.6	0.009
Feed intake/day (g)	93.2	94.4	92.2	88.5	3.52	0.677
Weight gain (g)	2426.2 ^a^	2215.3 ^b^	2438.9 ^a^	2345.6 ^a^	36.6	0.009
Average daily gain (g/d)	57.8 ^a^	52.7 ^b^	58.1 ^a^	55.8 ^a^	0.87	0.009
Feed conversion ratio	1.6	1.8	1.6	1.6	0.08	0.252
Mortality %	0	6.7	0	3.3		

^1^ B—basal diet (corn and soybean-based); B + S—basal diet + *Salmonella typhimurium* challenged; B + P—basal diet + *P. pentosaceus* − not *Salmonella typhimurium* challenged; B + P + S—basal diet + *P. pentosaceus* + *Salmonella typhimurium* challenged. ^2^ SEM—standard error of mean. ^a,b^ Means in the same row with different superscripts differ significantly (*p* < 0.05). Means in the same row without superscripts are similar (*p* > 0.05).

**Table 3 animals-14-01676-t003:** Serum biochemistry activities of broilers fed with *Pediococcus pentosaceus* GT001 (n = 8).

Items ^2^	Treatments ^1^				SEM ^3^	*p*-Value
	B	B + S	B + P	B + P + S		
**Liver function**						
Albumin (g/L)	14.3 ^a^	9.9 ^b^	15.4 ^a^	10.9 ^b^	0.78	<0.001
Globulin (g/L)	19.4 ^ab^	14.5 ^b^	20.6 ^a^	14.2 ^b^	1.29	0.004
Creatinine (ummol/L)	27.5 ^a^	17.9 ^c^	27.1 ^a^	21.4 ^b^	0.98	<0.001
Total protein (g/L)	32.9 ^ab^	24.6 ^c^	34.3 ^a^	28.1 ^bc^	1.46	0.001
ALT (U/L)	14.4 ^b^	9.8 ^c^	18.9 ^a^	16.6 ^ab^	0.81	<0.001
AST (U/L)	177.6 ^ab^	152.2 ^c^	181.1 ^a^	169.5 ^b^	2.24	<0.001
**Lipid profile (mmol/L)**						
T. cholesterol	282.6 ^a^	223.6 ^b^	139.8 ^c^	163.5 ^c^	16.0	<0.001
TG	69.8 ^a^	64.1 ^a^	48.9 ^b^	48.1 ^b^	2.78	<0.001
HDL	65.2 ^a^	63.1 ^a^	40.7 ^b^	38.5 ^b^	2.97	<0.001
LDL	168.3 ^b^	157.0 ^b^	277.2 ^a^	250.4 ^a^	15.0	<0.001

^1^ B—basal diet (corn and soybean-based); B + S—basal diet + *Salmonella typhimurium* challenged; B + P—basal diet + *P. pentosaceus* − not *Salmonella typhimurium* challenged; B + P + S—basal diet + *P. pentosaceus* + *Salmonella typhimurium* challenged. ^2^ ALT—alanine aminotransferase AST—aspartate aminotransferase TG—triglycerides; HDL—high-density lipoprotein; LDL—low-density lipoprotein; T. cholesterol—total cholesterol. ^3^ SEM—standard error of mean. ^a–c^ Means in the same row with different superscripts differ significantly (*p* < 0.05).

**Table 4 animals-14-01676-t004:** *Pediococcus pentosaceus* GT001 impact on the serum antioxidant activities of broilers (n = 8).

Items ^2^	Treatments ^1^				SEM ^3^	*p*-Value
	B	B + S	B + P	B + P + S		
T-AOC (nmol/L)	0.53 ^ab^	0.45 ^b^	0.60 ^a^	0.48 ^b^	0.02	0.001
SOD (nmol/L)	138.1 ^b^	127.7 ^c^	146.8 ^a^	137.0 ^b^	1.58	<0.001
MDA (U/mL)	6.7 ^a^	6.1 ^a^	4.9 ^b^	5.2 ^b^	0.22	<0.001
CAT (U/mL)	322.6 ^c^	329.8 ^c^	391.7 ^a^	348.5 ^b^	4.15	<0.001
GHS-Px (U/mL)	539.4 ^b^	525.6 ^c^	592.2 ^a^	563.6 ^b^	7.02	<0.001

^1^ B—basal diet (corn and soybean-based); B + S—basal diet + *Salmonella typhimurium* challenged; B + P—basal diet + *P. pentosaceus* − not *Salmonella typhimurium* challenged; B + P + S—basal diet + *P. pentosaceus* + *Salmonella*. ^2^ T-AOC—total antioxidant capacity; SOD—superoxide dismutase; MDA—malondialdehyde; CAT—catalase; GHS-Px glutathione peroxidase. ^3^ SEM—standard error of mean. ^a–c^ Means in the same row with different superscripts differ significantly (*p* < 0.05).

**Table 5 animals-14-01676-t005:** Effects of probiotic *Pediococcus pentosaceus* GT001 on the serum cytokines and immunological indices of broilers (n = 8).

Items ^2^	Treatments ^1^				SEM ^3^	*p*-Value
	B	B + S	B + P	B + P + S		
**Cytokines (pg/mL)**						
TNF-α	119.2 ^ab^	103.9 ^c^	115.9 ^a^	111.8 ^b^	1.70	<0.01
IL 6	65.4 ^ab^	57.4 ^c^	63.6 ^a^	61.8 ^b^	0.63	<0.01
IL 10	31.3 ^b^	30.3 ^b^	35.6 ^a^	32.3 ^b^	0.76	<0.01
**Immunology (g/L)**						
IgA	1.2 ^a^	0.9 ^b^	1.3 ^a^	1.2 ^a^	0.05	0.001
IgG	8.9 ^a^	8.3 ^b^	9.2 ^a^	8.8 ^a^	0.10	<0.01
IgM	0.97 ^ab^	0.83 ^c^	1.04 ^a^	0.91 ^b^	0.02	<0.01

^1^ B—basal diet (corn and soybean-based); B + S—basal diet + *Salmonella typhimurium* challenged; B + P—basal diet + *P. pentosaceus* − not *Salmonella typhimurium* challenged; B + P + S—basal diet + *P. pentosaceus* + *Salmonella typhimurium* challenged. ^2^ TNF-α—tumor necrosis factor-alpha. IL 6—interleukin 6. IL 10—interleukin 10. IgA—immunoglobulin A. IgG—immunoglobulin G. IgM—immunoglobulin M. ^3^ SEM—standard error of mean. ^a–c^ Means in the same row with different superscripts differ significantly (*p* < 0.05).

**Table 6 animals-14-01676-t006:** Effect of *Pediococcus pentosaceus* GT001 on the digestive enzymes of broilers (n = 8).

Items (ng/mL)	Treatments ^1^				SEM ^2^	*p*-Value
	B	B + S	B + P	B + P + S		
**Serum**						
Amylase	54.7 ^b^	46.9 ^c^	63.1 ^a^	61.1 ^a^	0.981	<0.01
Lipase	29.7 ^b^	27.9 ^b^	40.8 ^a^	38.0 ^a^	1.47	<0.01
**Intestine**						
Amylase	24.1 ^b^	21.5 ^c^	27.3 ^a^	23.3 ^bc^	0.529	<0.01
Lipase	12.9 ^b^	11.4 ^c^	14.7 ^a^	13.6 ^ab^	0.336	<0.01

^1^ B—basal diet (corn and soybean-based); B + S—basal diet + *Salmonella typhimurium* challenged; B + P—basal diet + *P. pentosaceus* − not *Salmonella typhimurium* challenged; B + P + S—basal diet + *P. pentosaceus* + *Salmonella typhimurium* challenged. ^2^ SEM—standard error of mean. ^a–c^ Means in the same row with different superscripts differ significantly (*p* < 0.05).

**Table 7 animals-14-01676-t007:** Effects of *Pediococcus pentosaceus* GT001 on the weights of the organs of broilers (n = 8).

Items (g)	Treatments ^1^				SEM ^2^	*p*-Value
	B	B + S	B + P	B + P + S		
Total viscera	251.5	237.4	244.9	255.7	9.39	0.548
Liver	44.2	38.4	39.6	40.6	2.46	0.406
Heart	9.3	8.7	8.8	9.3	0.43	0.636
Kidney	10.1	10.5	10.6	8.8	0.65	0.232
Gizzard	52.9	47.7	54.7	50.8	2.11	0.153
Empty gizzard	36.4	35.3	38.8	37.4	1.53	0.429
Spleen	4.1 ^ab^	2.2 ^c^	5.3 ^a^	3.5 ^bc^	0.34	<0.01
Bursa	4.0 ^a^	2.3 ^b^	4.6 ^a^	4.6 ^a^	0.47	0.009
Pancreas	4.7	4.1	4.4	4.9	0.25	0.099
Lungs	11.1	9.7	10.7	9.6	0.52	0.156
Small intestine	62.1 ^ab^	55.2 ^b^	60.5 ^a^	68.4 ^a^	2.93	0.043

^1^ B—basal diet (corn and soybean-based); B + S—basal diet + *Salmonella typhimurium* challenged; B + P—basal diet + *P. pentosaceus* − not *Salmonella typhimurium* challenged; B + P + S—basal diet + *P. pentosaceus* + *Salmonella typhimurium* challenged. ^2^ SEM—standard error of mean. ^a–c^ Means in the same row with different superscripts differ significantly (*p* < 0.05).

**Table 8 animals-14-01676-t008:** Effect of *Pediococcus pentosaceus* GT001 on broiler meat quality.

Items ^2^	Treatments ^1^				SEM ^3^	*p*-Value
	B	B + S	B + P	B + P + S		
Color L*	51.4	49.6	52.1	49.3	0.962	0.156
a*	10.2	9.2	10.1	9.3	0.538	0.494
b*	8.9	9.3	9.1	8.2	0.409	0.299
Breast pH	5.9	5.7	6.8	5.7	0.103	0.429
Drip loss %	11.3	11.9	10.1	10.9	1.11	0.723

^1^ B—basal diet (corn and soybean-based); B + S—basal diet + *Salmonella typhimurium* challenged; B + P—basal diet + *P. pentosaceus* − not *Salmonella typhimurium* challenged; B + P + S—basal diet + *P. pentosaceus* + *Salmonella typhimurium* challenged. ^2^ L*—lightness; a*—redness; b*—yellowness. ^3^ SEM—standard error of mean.

**Table 9 animals-14-01676-t009:** Effect of *Pediococcus pentosaceus* GT001 on the intestinal length and pH of broilers (n = 8).

Items	Treatments ^1^				SEM ^2^	*p*-Value
	B	B + S	B + P	B + P + S		
**Length (cm)**						
Duodenum	30.8	26.5	31.6	29.0	2.11	0.359
Jejunum	77.6	73.2	77.4	75.0	4.44	0.065
Ileum	73.0	61.5	71.4	68.6	2.94	0.062
**pH**						
Duodenum	6.0 ^ab^	5.8 ^c^	6.2 ^a^	5.9 ^bc^	0.06	<0.01
Jejunum	6.7 ^a^	6.4 ^b^	6.2 ^b^	6.2 ^b^	0.05	<0.01
Ileum	6.9 ^a^	6.7 ^b^	6.4 ^bc^	6.2 ^c^	0.07	<0.01

^1^ B—basal diet (corn and soybean-based); B + S—basal diet + *Salmonella typhimurium* challenged; B + P—basal diet + *P. pentosaceus* − not *Salmonella typhimurium* challenged; B + P + S—basal diet + *P. pentosaceus* + *Salmonella typhimurium* challenged. ^2^ SEM—standard error of mean. ^a–c^ Means in the same row with different superscripts differ significantly (*p* < 0.05).

**Table 10 animals-14-01676-t010:** Effect of *Pediococcus pentosaceus* GT001 on the small intestinal morphology of broilers.

Items ^2^	Treatments ^1^				SEM ^3^	*p*-Value
	B	B + S	B + P	B + P + S		
**Duodenum (μm)**						
VH	1137.5 ^b^	991.3 ^c^	1258.5 ^a^	1114.5 ^b^	19.0	<0.01
CD	190.2 ^a^	165.2 ^b^	202.2 ^a^	187.8 ^a^	5.24	0.001
VH:CD	6.0	6.0	6.2	6.0	0.20	0.770
**Jejunum (μm)**						
VH	1046.1 ^a^	820.0 ^c^	942.6 ^b^	952.2 ^b^	16.7	<0.001
CD	182.4 ^b^	164.3 ^c^	203.6 ^a^	185.1 ^b^	3.25	<0.001
VH:CD	5.2	5.0	5.1	5.2	0.10	0.584
**Ileum (μm)**						
VH	764.7 ^b^	632.7 ^c^	857.5 ^a^	774.3 ^b^	19.6	<0.001
CD	151.9 ^b^	124.9 ^c^	166.0 ^a^	150.4 ^b^	3.35	<0.001
VH:CD	5.0	5.1	5.2	5.2	0.18	0.917

^1^ B—basal diet (corn and soybean-based); B + S—basal diet + *Salmonella typhimurium* challenged; B + P—basal diet + *P. pentosaceus* − not *Salmonella typhimurium* challenged; B + P + S—basal diet + *P. pentosaceus* + *Salmonella typhimurium* challenged. ^2^ VH—villus height; CD—crypt depth; VH:CD—villus height and crypt depth ratio. ^3^ SEM—standard error of mean. ^a–c^ Means in the same row with different superscripts differ significantly (*p* < 0.05). Means in the same row without superscripts are similar (*p* > 0.05).

**Table 11 animals-14-01676-t011:** Effect of *Pediococcus pentosaceus* GT001 on the *Salmonella* load after *Salmonella* infection in broilers.

Items ^2^	Treatments ^1^				SEM ^3^	*p*-Value
	B	B + S	B + P	B + P + S		
***Salmonella*** (log 10^4^)						
Day 3 PI	0 ^b^	4.56 ^a^	0 ^b^	0.0015 ^b^	1722	<0.01
Day 7 PI	0 ^b^	4.30 ^a^	0 ^b^	0.0005 ^b^	1414	<0.01
Day 14 PI	0 ^b^	3.92 ^a^	0 ^b^	0 ^b^	1279	<0.01
**TVC** (log 10^4^)						<0.01
Day 3 PI	10.50 ^c^	20.62 ^b^	21.12 ^b^	28.70 ^a^	5108	<0.01
Day 7 PI	18.50 ^a^	20.80 ^a^	3.64 ^b^	3.10 ^b^	18,906	<0.01
Day 14 PI	39.80 ^a^	22.60 ^b^	3.10 ^c^	2.76 ^c^	7794	<0.01

^1^ B—basal diet (corn and soybean-based); B + S—basal diet + *Salmonella typhimurium* challenged; B + P—basal diet + *P. pentosaceus* − not *Salmonella typhimurium* challenged; B + P + S—basal diet + *P. pentosaceus* + *Salmonella typhimurium* challenged. ^2^ TVC—total viable count; PI—post infection. ^3^ SEM—standard error of mean. ^a–c^ Means in the same row with different superscripts differ significantly (*p* < 0.05). Means in the same row without superscripts are similar (*p* > 0.05).

**Table 12 animals-14-01676-t012:** Dietary inclusion of *Pediococcus pentosaceus* GT001 on the composition of the cecal microbiota of broilers by alpha diversity measures.

Item	Treatments ^1^				SEM ^2^	*p*-Value
	B	B + S	B + P	B + P + S		
Goods coverage	1	1	1	1	0.002	0.96
Shannon	7.06	7.01	7.07	7.03	0.119	0.990
Simpson	0.97	0.97	0.98	0.97	0.005	0.867
Chao1	562.7	526.8	568.8	594.2	32.6	0.549
Observed OTUs	566.7	527.2	591.5	549.5	31.6	0.539

^1^ B—basal diet (corn and soybean-based); B + S—basal diet + *Salmonella typhimurium* challenged; B + P—basal diet + *P. pentosaceus* − not *Salmonella typhimurium* challenged; B + P + S—basal diet + *P. pentosaceus* + *Salmonella typhimurium* challenged. ^2^ SEM—standard error of mean. Means in the same row without superscripts are similar (*p* > 0.05).

**Table 13 animals-14-01676-t013:** Effect of dietary inclusion of *Pediococcus pentosaceus* GT001 on the bacterial taxonomy within the composition of the cecum microbiota of broiler chickens.

Item	Treatments ^1^				SEM ^2^	*p*-Value
	Relative Abundance (%)					
	B	B + S	B + P	B + P + S		
Phylum						
*Cyanobacteria*	0.07 ^a^	0.01 ^b^	0.09 ^a^	0.02 ^b^	0.02	0.033
*Actinobacteria*	0.07 ^b^	0.05 ^b^	0.07 ^b^	0.15 ^a^	0.02	0.002
*Proteobacteria*	3.88	2.78	3.39	3.21	0.58	0.615
Family						
*Peptostreptococcaceae*	0.10	0.10	0.30	0.20	0.116	0.139
*Bacteroidaceae*	5.24	3.87	4.20	4.31	1.19	0.504
*Clostridiaceae*	0.01 ^b^	0.01 ^b^	0.05 ^a^	0.01 ^b^	0.014	0.054
*Lactobacillaceae*	4.96 ^a^	2.43 ^b^	5.24 ^a^	2.76 ^b^	0.688	0.010
Genus						
*Clostridiales_unclassified*	1.67	1.27	1.69	1.32	0.191	0.297
*Lactobacillus*	4.87 ^a^	2.39 ^b^	5.01 ^a^	2.51 ^b^	0.678	0.010
*Anaerofustis*	0.005	0.001	0.006	0.004	0.002	0.211
*Escherichia-Shigella*	0.51 ^a^	0.13 ^b^	0.13 ^b^	0.13 ^b^	0.120	0.006
*CHKCI001*	0.50 ^a^	0.20 ^b^	0.30 ^b^	0.21 ^b^	0.08	0.044
*CHKCI002*	0.03	0.01	0.02	0.02	0.007	0.463
*Papillibacter*	0.03	0.02	0.04	0.03	0.007	0.269
*GCA-900066225*	0.41	0.28	0.37	0.36	0.113	0.210
*GCA-900066575*	0.87 ^a^	0.53 ^b^	0.75 ^ab^	0.61 ^b^	0.082	0.044

^1^ B—basal diet (corn and soybean-based); B + S—basal diet + *Salmonella typhimurium* challenged; B + P—basal diet + *P. pentosaceus* − not *Salmonella typhimurium* challenged; B + P + S—basal diet + *P. pentosaceus* + *Salmonella typhimurium* challenged. ^2^ SEM—standard error of mean. ^a,b^ Means in the same row with different superscripts differ significantly (*p* < 0.05). means in the same row without superscripts are similar (*p* > 0.05).

## Data Availability

Upon request, the corresponding author may supply the data supporting the study’s findings.

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
