# Peer review of "Influence of Pediococcus pentosaceus GT001 on Performance, Meat Quality, Immune Function, Antioxidant and Cecum Microbial in Broiler Chickens Challenged by Salmonella typhimurium"

_animals, 2024, doi:10.3390/ani14111676_

Round 1
Reviewer 1 Report
Comments and Suggestions for Authors
In the manuscript, the authors studied the Influence of Pediococcus pentosaceus GT001 on Performance, Meat quality, Immune function, Antioxidant and Cecum microbial in Broiler Challenged by Salmonella typhimurium.
Notes.
1. Line 33 (abstract). Two hundred day old broiler chickens? There is most likely a mistake here! There are no such growing periods. Moreover, these are no longer chickens.
2. Materials and research methods. Why did they use different concentrations of the pathogen during infection in diets? In particular, for the same pathogen strain?
3. The abstract does not compare the results obtained with the control group! Of significant interest is the analysis not only between the experimental groups, but also regarding the control.
4. Line 93. How to uniformly mix the inoculum with the diet feed. How applicable is this technology in an industrial environment?
5. Line 95. What does sexless chickens mean? You need to indicate what gender they were. And that gender was not taken into account in the study.
6. Conclusion. Edit. More details and research results.
Author Response
Section B: Responses to the comments by reviewer 2
General comment: In the manuscript, the authors studied the Influence of Pediococcus pentosaceus GT001 on Performance, Meat quality, Immune function, Antioxidant and Cecum microbial in Broiler Challenged by Salmonella typhimurium.
Authors response: We appreciate the reviewer's insightful remarks. We have implemented the changes based on the recommendations. A revised manuscript with the detailed modifications have been highlighted.
Comment 1: Line 33 (abstract). Two hundred day old broiler chickens? There is most likely a mistake here! There are no such growing periods. Moreover, these are no longer chickens
Authors’ response: The sentence has been modified (Please see line 33).
Comment 2: Materials and research methods. Why did they use different concentrations of the pathogen during infection in diets? In particular, for the same pathogen strain?
Authors’ response: The pathogen used during infection was same concentration for the challenged groups. (Please see lines 38 and 122).
Comment 3: The abstract does not compare the results obtained with the control group! Of significant interest is the analysis not only between the experimental groups, but also regarding the control.
Authors’ response: There has been a little modification in the abstract comparing the results with the control group. (Please see lines 43, 45 and 46. It’s been highlighted in red).
Comment 4: Line 93. How to uniformly mix the inoculum with the diet feed. How applicable is this technology in an industrial environment?
Authors’ response: The uniform mixing of inoculum in feed is highly feasible in an industrial environment due to the availability of advanced mixing equipment, scalable processes and the potential for automation. Ensuring consistent, high-quality mixtures can enhance product quality and operational efficiency, making this technology a valuable asset for large-scale production facilities.
Comment 5: Line 95. What does sexless chickens mean? You need to indicate what gender they were. And that gender was not taken into account in the study.
Authors’ response: Gender was not taken into account in the study and therefore the statement has been modified. (Please see line 107).
Comment 6: Conclusion. Edit. More details and research results.
Authors’ response: There has been a little modification. (Please see lines 721-728. It has been highlighted red).

Reviewer 2 Report
Comments and Suggestions for Authors
The manuscript provides interesting information in the field of use of feed additives like probiotics on broiler chicken production. In my opinion, only some minor revisions to the text are necessary.
Throughout the manuscript:
When you write: „higher”, „lower”, „similar”, etc., please indicate the significance level. You can omit the word "significantly" when you indicate the significance level.
All Latin names and terms should be italicized.
Please insert a space between number and unit.
Please consistently describe the level of significance as „P <0.05)” or „p < 0.05)”.
Keywords
I propose the following order: poultry, broiler, Salmonella typhimurium, probiotic, Pediococcus pentosaceus.
Introduction
Please add more information about P. pentosaceus.
Materials and Methods
Lines 172-175: Please insert the references.
Lines 175-177: Please insert the references.
Lines 175-177: L*, a*, b* -please write „*” in superscript.
Results
Tables 10 and 11: Please avoid units in the "Items" column title.
Discussion
Lines 507-508: I suggest remove.
Line 527: I suggest replace „In accordance with our study, de Azevedo et al. [18] ...” with „Our results were accordance to results of study performed by [18] ...”.
Lines 569-570: I suggest remove.
Line 590: I suggest remove. References "[34-36]" may be placed at the end of the following sentence.
Line 607: Please insert the references at the end of the sentence.
Lines 613-615: I suggest remove.
Line 619: Please insert the references after “Research”.
Line 627: Please insert the references after “researchers”.
Lines 640-647: I suggest remove.
Line 661: Please insert a space.
Conclusions
The second sentence should be written as follows: Dietary inclusion of P. pentosaceus GT001 as probiotic led to significant changes in growth per formance, immune function, digestive enzymes and intestinal morphology of Salmonella typhimurium challenged broiler chickens.
Line 709: Please inser a dot.
References
Please check the spelling of journal names and their abbreviations.
Author Response
Section C: Responses to the comments by reviewer 3
General comment: The manuscript provides interesting information in the field of use of feed additives like probiotics on broiler chicken production. In my opinion, only some minor revisions to the text are necessary.
Authors response: We appreciate the reviewer's insightful remarks and recommendations. We have implemented extensive changes based on the recommendations. A revised manuscript with the detailed modifications have been highlighted.
Comment 1: When you write: „higher”, „lower”, „similar”, etc., please indicate the significance level. You can omit the word "significantly" when you indicate the significance level.
Authors’ response: All significantly have been omitted and level of significance indicated as suggested. (Please see lines 246, 262, 264, 267, 269, 270, 272, 273, 285, 288, 290, 291, 292, 303, 304, 310, 322, 324, 329, 340, 343, 344, 381, 382, 386, 387, 389, 400, 401, 402, 404, 409, 410, 411, 423, 428, 492, 493, 494, 495, 496, 497 and 498)
Comment 2: All Latin names and terms should be italicized.
Authors’ response: All Latin names have been italicized. Thank you. (Kindly see line 69, 433,513,539,667,668 and 696 in the revised version. It’s been highlighted red).
Comment 3 Please insert a space between number and unit.
Authors’ response: It has been modified. (Kindly see line 108,120, in the revised version. It’s been highlighted in red).
Comment 4: Please consistently describe the level of significance as „P <0.05)” or „p < 0.05)”.
Authors’ response: All significant levels have been revised and are consistent now. Thank you. (Kindly see lines 246, 247 and in the revised version. It’s been highlighted in red).
Comment 5: I propose the following order: poultry, broiler, Salmonella typhimurium, probiotic, Pediococcus pentosaceus.
Authors’ response: As suggested, the keywords have been modified. Thank you. (Kindly see lines 57and 58. It’s been highlighted in red).
Comment 6: Please add more information about P. pentosaceus.
Authors’ response: More information on P. pentosaceus has been added as suggested. Thank you. (Kindly see lines 83-89 and 91-93 In the revised version. It’s been highlighted in red).
Comment 7: Lines 172-175: Please insert the references.
Authors’ response: The reference has been inserted. (Please see line 187. It has been highlighted in red)
Comment 8: Lines 175-177: Please insert the references.
Authors’ response: The reference has been inserted. (Please see line 189. It has been highlighted in red)
Comment 9: Lines 175-177: L*, a*, b* -please write „*” in superscript.
Authors’ response: it has been modified in the manuscript as requested. (Kindly see lines 188 and 189 in the revised version. It’s been highlighted in red).
Comment 10: Tables 10 and 11: Please avoid units in the "Items" column title.
Authors’ response: Table 10 and 11 have been modified in the revised version. It’s been highlighted in red).
Comment 11: Lines 507-508: I suggest remove.
Authors’ response: It has been removed as suggested. Thank you. (kindly see line 523 and 524. It has been highlighted red).
Comment 12: Line 527: I suggest replace „In accordance with our study, de Azevedo et al. [18] ...” with „Our results were accordance to results of study performed by [18] ...”.
Authors’ response: It has been modified as suggested. Thank you. (Kindly see line 543 and 544. It has been highlighted red).
Comment 13: Lines 569-570: I suggest remove.
Authors’ response: It has been removed as suggested. (Kindly see line 607 in the revised version. It’s been highlighted in red).
Comment 14: Line 590: I suggest remove. References "[34-36]" may be placed at the end of the following sentence.
Authors’ response: It has been modified as suggested. (Please see lines 608-609. It has been highlighted red).
Comment 15: Line 607: Please insert the references at the end of the sentence.
Authors’ response: Reference inserted. (Please see line 626. It has been highlighted red)
Comment 16: Lines 613-615: I suggest remove.
Authors’ response: Removed as suggested. (Please see lines 632-634. It has been highlighted red).
Comment 17: Line 619: Please insert the references after “Research”.
Authors’ response: It has been modified as suggested. (Please see line 638. It’s been highlighted red).
Comment 18: Line 627: Please insert the references after “researchers”.
Authors’ response: the references have been inserted after the researchers as suggested. Thank you. (Kindly see line 647. It’s been highlighted red).
Comment 19: Lines 640-647: I suggest remove.
Authors’ response: It has been removed as suggested. (Kindly see lines 659-666. It’s been highlighted red).
Comment 20: Line 661: Please insert a space.
Authors’ response: Space has been inserted. (Please see line 680. It’s been highlighted red)
Comment 21: The second sentence should be written as follows: Dietary inclusion of P. pentosaceus GT001 as probiotic led to significant changes in growth performance, immune function, digestive enzymes and intestinal morphology of Salmonella typhimurium challenged broiler chickens.
Authors’ response: The second sentence has been modified as suggested. (Kindly see lines 721-724. It’s been highlighted red).
Comment 22: Line 709: Please insert a dot.
Authors’ response: The dot has been inserted. Please see line 731.
Comment 23: Please check the spelling of journal names and their abbreviations.
Authors’ response: Checked. Thank you